# A Non-Destructive Method, Micro-CT, Supports the Identification of Three New *Casmara* Species from Sumatra and Taiwan (Lepidoptera: Ashinagidae) [note 1]

**DOI:** 10.3390/insects16080747

**Published:** 2025-07-22

**Authors:** In-Won Jeong, Sora Kim, John B. Heppner

**Affiliations:** 1Laboratory of Insect Phylogenetics and Evolution, Department of Plant Protection & Quarantine, Jeonbuk National University, Jeonju 54896, Republic of Korea; scott9778@jbnu.ac.kr; 2Department of Agricultural Convergence Technology, Jeonbuk National University, Jeonju 54896, Republic of Korea; 3McGuire Center for Lepidoptera and Biodiversity, Florida Museum of Natural History, University of Florida, Gainesville, FL 32611, USA; jheppner@flmnh.ufl.edu

**Keywords:** Ashinagidae, *Casmara*, new species, Micro-CT

## Abstract

Many insects, including Lepidoptera, exhibit high morphological similarity among species, so identifying them typically requires dissecting the abdomen to examine genitalia structures. However, this process often damages specimens and flattens their original three-dimensional features. To address these issues, this study tested a non-destructive method, micro-computed tomography, which used X-rays to visualize internal structures without harming specimens. Applying this technique to the genus *Casmara*, which has had no new species reported for nearly a decade and lacks follow-up research since its taxonomic revision, we discovered three new species from Sumatra and Taiwan. By comparing the results obtained from this method with traditional dissections, we confirmed that the new technique significantly enhances species identification. Digital models of male genitalia produced by this method can be freely rotated, magnified, and even reproduced through three-dimensional printing. Additionally, this study updated the global species list and distribution records for the genus. This approach preserves the integrity of specimens and allows detailed internal observations, thereby contributing significantly to faster, safer species identification and improved sharing of research data.

## 1. Introduction

Insects, arthropods characterized by having six legs, represent the most diverse and successful group of organisms, accounting for more than half of all described species. These organisms provide significant ecological services such as pollination, decomposition, and maintenance of wildlife [1,2]. Despite this diversity, external insect morphology can be similar across different species, necessitating examination of genitalia structures for accurate species identification [3]. However, this technique necessitates destructive sampling of holotypes for new species, posing a risk of damaging specimens during the separation and mounting of the genitalia. Moreover, flattening an inherently 3D (three-dimensional) structure into 2D (two dimensions) often leads to the loss of essential morphological details.

To overcome these limitations, Micro-CT (Computed Tomography), which allows for the non-destructive 3D visualization of internal structures, is becoming an increasingly valuable tool in insect taxonomy, including Lepidoptera [4,5,6]. Micro-CT was first developed by Elliot and Dover [7], and the basic principle involves acquiring X-ray images from multiple angles—either by rotating the sample or moving the scanner around it—and then reconstructing a 3D model from these projections [8]. In addition to generating detailed 3D reconstructions, this technique enables the virtual rotation, magnification, and sectioning of specimens in a digital environment. Due to these advantages, Micro-CT has found widespread application in areas as varied as enabling rapid and accurate cancer diagnosis [9], non-invasively examining ancient papyri without physically unrolling them [10], and identifying internal morphological features in fossils [11].

In this study, we use Micro-CT to investigate the male genitalia of three new species in the genus *Casmara* Walker, 1863, collected from the Oriental realm. We also present traditional dissection images alongside the Micro-CT data to compare both methods and evaluate whether Micro-CT can facilitate the reliable identification of new species.

The genus *Casmara* was reclassified from Oecophoridae to Stathmopodidae through a molecular phylogenetic study by Kim, Kaila, and Lee [12] using eight genes, including *C*. *agronoma*. Thereafter, based on phylogenetic analyses using eight genes, including *C*. *patrona*, the genus was ultimately reclassified into Ashinagidae [13]. However, subsequent studies on this genus have not been conducted; the last new species were those described by Lvovsky [14] from Southeast Asia, and no further species have been reported for over a decade. Table 1 lists the species of *Casmara* reported to date, including the new species described in this study and their distributions. As a result, knowledge of this genus remains limited, underscoring the need for additional studies. Furthermore, 21 species in this genus were documented to bore into stems for feeding and potentially act as serious agricultural and forestry pests; therefore, it is essential to establish accurate species identification and distribution [15]. Consequently, this study not only expands the diversity of the genus *Casmara* by reporting new species but also provides a 3D structure of the genitalia using Micro-CT, offering previously unknown details.

## 2. Materials and Methods

The publication and its nomenclatural acts have been registered in ZooBank, and the LSID (Life Science Identifiers) for this publication is as follows: urn:lsid:zoobank.org:pub: B7BC9C0F-60BD-44EF-8AA3-74EEC200291D. Adults and dissected genitalia were photographed using a Leica Z16APO (Leica, Wetzlar, Germany) equipped with a Dome Illuminator Leica LED5000HDI (Leica, Wetzlar, Germany), with image capture performed using the Mosaic software (v. 2.4, Tucsen, Fuzhou, China). Helicon Focus software (v. 8.2.2 Pro, Helicon soft, Kharkov, Ukraine) was used to capture images at multiple focal planes and merge them into a single composite image. Adobe Photoshop 2024 (Adobe, San Jose, CA, USA) was used to remove unnecessary parts and backgrounds from the merged images. The terminology used for morphology and genitalia structures follows Heppner [26]. The methods and terminology for Micro-CT imaging follow Latief, Sari, and Fitri [27]. Holotypes are deposited with the McGuire Center for Lepidoptera and Biodiversity (MGCL), Gainesville, FL, USA.

### 2.1. Specimen Preparation

The abdomen was separated from the specimen because the metal pin at the thorax causes photon starvation and beam hardening, which can result in dark or bright streaks that may interfere with image generation [28]. The separated abdomen was placed inside a Snap-Fit Gelatin^®^ Capsule (Agar Scientific, Rotherham, UK) with adhesive applied to the cephalic margin and suspended in the center to minimize contact with surrounding materials, thereby reducing noise.

### 2.2. Specimen Scanning

Scanning was performed using the Skyscan 1276 (Bruker, Kontich, Belgium); additionally, the positions and reconstruction parameters were set using the NRecon software (v. 2.0.0.5, Bruker, Kontich, Belgium) (Figure 1). Other parameters are provided in Table 2.

### 2.3. Genitalia Dissection

Genitalia were dissected from the abdomen, and specimens were prepared according to the method of Kim, Lee, and Lee [29]. Microstructures that could not be observed using Micro-CT were photographed using a Leica MSV266 (Leica, Wetzlar, Germany) with Optiview software ver. X64 (Korea Lab Tech, Seongnam, Republic of Korea) and presented in the dissected view.

### 2.4. 3D Reconstruction of Genitalia

Micro-CT datasets were processed with DataViewer (v. 1.5.6.2), CTAn (v. 1.20.3.0), and CTvox (v. 3.3.1) (Bruker, Kontich, Belgium). Initial inspection and orientation were performed in DataViewer, where orthogonal X-Y, Y-X, and Z-X slices were examined and merged into an ortho-slice to verify sample integrity and to define a preliminary VOI (volume of interest) (Figure 2). The refined ROI (region of interest) was subsequently delineated in CTAn to isolate the genitalia; abdominal segments and debris were excluded, and the segmented volume was exported as an STL file for downstream 3D analysis and printing (Figure 3). Finally, CTvox was used for volumetric rendering: the segmented model was interactively rotated, sliced, and annotated in a virtual environment, and opacity and brightness were adjusted to emphasize internal structures.

## 3. Results

Order Lepidoptera Linnaeus, 1758Family Ashinagidae Matsumura, 1929Genus *Casmara* Walker, 1863 [23]Type species: *Casmara infaustella* Walker, 1863


***Casmara falcatussica* sp. nov.**


urn:lsid:zoobank.org:act:C609D314-8AE8-4A7E-98D3-F7779305741B

MorphoSource DOI of Micro-CT images: https://doi.org/10.17602/M2/M743635

MorphoSource DOI of 3D genitalia: https://doi.org/10.17602/M2/M743940

**Diagnosis.** This species is superficially similar to *C*. *longiclavata* but has differences in the wing markings. In *C*. *longiclavata*, the forewing has a dark gray ground color with distinct markings, and the apex is dark brown except for the area along the veins. The hindwing has a grayish ocher ground color with veins and posterior margin from the base to the middle gray. But in this species, the forewing has ocherous ground color with indistinct markings, and the apex is entirely brown. The hindwing is entirely pale ocher, with ocherous veins. In the male genitalia, this species is similar to *C*. *longiclavata*, but it can be distinguished by the gnathos and aedeagus. In *C*. *longiclavata*, the structure near the apex of the gnathos is bar-shaped with an oblique apex in lateral view. The cornutus extends beyond the apex of the aedeagus and is curved from 1/2 to the apex, with an obliquely connected triangular process at 2/5. In this species, the structure near the apex of the gnathos is cone-shaped with a slightly curved apex in lateral view. The cornutus is straight, not extending to the apex of the aedeagus, bearing a microspine at the base and a triangular process connected vertically at 1/3. This species is also similar to *C*. *acantha*, but it differs in the structure of the cucullus and aedeagus. In *C*. *anatha*, the cucullus is a blunt triangular, with a sharp bend at the middle of the ventral margin. The cornutus is 1/4 of the length of aedeagus and slightly curved at the apex. But in this species, the cucullus is narrow and triangular with a gentle curve, and the cornutus is 1/3 the length of aedeagus and entirely straight.

**Description. Adult** (Figure 4). *Head*. Frons and vertex ocher; occiput ocher, gradually shifting to pale brown toward the apex. Antenna pale ocher with cilia; scape pale brown from base to 2/3. Labial palpus pale ocher; first segment dark brown dorsally; second segment irregular mixture of dark brown and brown; third segment brown ventrally. *Thorax*. Tegula brown, paler toward the apex with a white edge. Thorax brown; prothorax brown, edges of central portion white, excluding the outermost quarters on both sides. *Abdomen*. Abdomen brown, pale ocherous scales at each segment caudally. *Wing*. Wing expanse 23.0 mm. Forewing from base to 1/5 and from 3/5 to 4/5 predominantly ocher, remaining sections predominantly dark brown; costal pale grayish brown from base to 3/7, grayish brown from 3/7 to 5/7; brown blotch along each margin of costal and posterior from base to 1/6; pale brown blotch from 1/8 to 1/4 at median; pale orangey brown blotchat 3/11; pale reddish brown blotch at 6/11; pale reddish brown blotch along costal margin from 9/14 to 6/7, not reaching costal; white streak at terminal line excluding venation part; dark grayish brown at apex; fringes short, base ocher gradually darker grayish brown toward the apex. Hindwing ocher; dark ocherous scales on venation; fringes grayish ocher. **Male genitalia** (Figure 5 and Figure 6). Uncus arrow-head shaped in dorsal view and curved downward in lateral view, lateral margin with some setae, pointed apex sharply bent downward. Gnathos as long as uncus, tub-shaped, base to 1/2 convex and 1/2 to apex concave at dorsal margin; microspines from 3/5 to apex, gradually larger toward apex; cone-shaped structure subtly bent at apex, about 1/5 length of gnathos, located at 8/9 of the ventral margin. Costa slightly convex, with a few short setae medially and a few long setae apically. Valva subtriangular with blunt apex; cucullus as long as uncus, curved triangular shape with long setae at the inner surface; harpe weakly developed, apex densely covered with long setae; sacculus with ventral margin gently convex, dorsal margin flat from near base to midpoint and abruptly concave from midpoint to apex, inner margin flat from base to 2/5 and gently concave thereafter, outer margin gently convex, spine-like setae aligned from 4/5 to apex, apex blunt. Aedeagus slightly longer than twice the length of uncus, sclerotized overall, once curved, gradually widening from 5/8 to 3/4, then narrowing there to apex; spines at 3/4 and 7/8 of ventral margin; cornutus approximately 1/3 length of aedeagus, at apex, narrow cone-shaped, with single microspine at base and triangular process vertically at 1/3. **Female unknown.**

**Remarks.** Micro-CT allowed for observation of spines on the ventral margin of the aedeagus, which were not visible in the dissected images. However, the microspine near the cornutus appeared indistinct.

**Etymology.** This species name is a composite word derived from the Latin, ‘falcatus’ and ‘sica’, meaning ‘sickle-shaped’ and ‘dagger’. This species name refers to the shape of the aedeagus, which resembles the kukri, a traditional weapon with the curved form of a Nepalese dagger.

**Type material.** Holotype: ♂, Taiwan, New Taipei City, Wulai district, 200 m, 17–19.vi. 1985, J. B. Heppner, gen. slide. No. JBNU IPE-13309/I.W. Jeong (deposited in MGCL).

**Distribution.** Taiwan: known only from the type locality.

***Casmara fulvacorona*** **sp. nov.**

urn:lsid:zoobank.org:act:CE579307-286D-4BDE-8876-5CDF01179516

MorphoSource DOI of Micro-CT images: https://doi.org/10.17602/M2/M743640

MorphoSource DOI of 3D genitalia: https://doi.org/10.17602/M2/M743943

**Diagnosis.** This species is similar to *C*. *aduncata* and *C*. *exculta*, but it differs in wing markings. In *C*. *aduncata*, the forewing is predominantly ocherous with a few irregularly scattered brown scales. In *C*. *exculta*, the forewing has a brown background, orange brown from 1/2 to 3/4, and the hindwing is entirely pale brown. However, in this species, the forewing has an ocherous background, brown from 1/4 to 2/3, and the hindwing is grayish brown with ocherous apex. In the male genitalia, this species is similar to *C*. *aduncata*, but it has differences in the cucullus and aedeagus. In *C*. *aduncata*, the dorsal side of the cucullus is concave from the base to the middle, becoming convex from the middle to the apex, and the apex of the aedeagus is curved. But in this species, the dorsal side of the cucullus is slightly concave, and the aedeagus is straight without a curved portion.

**Description. Adult** (Figure 7). *Head*. Frons and vertex pale ocher; occiput ocher at base. Antenna pale ocher with cilia; scape dark brown at base; each flagellomere from 1/12 to 2/3 ocher with grayish brown apex. Labial palpus pale ocher; first segment grayish brown dorsally; second segment irregularly mixed with grayish brown and ocher scales; third segment grayish brown at 2/3 with a blunt apex. *Thorax*. Tegula pale ocher, inner part of center pale brown. Thorax pale ocher; prothorax pale ocher, with the area corresponding to 1/3 from each and composed of pale ocherous scales with brown apex; mesothorax with symmetrical brown blotches at the cephalic margin. *Abdomen*. Abdomen brownish black; first segment with a pale ocherous triangular spot at the center of caudal margin; second segment with an obscure ocherous spot at the center of caudal margin. *Wing*. Wing expanse 48.6–50.1 mm. Forewing primarily ocher from base to 1/4, brown and grayish brown mixed from 1/4 to apex and gradually dispersing toward apex; costal ocher; base grayish brown; first fascia brown from base to 2/17; second fascia grayish brown, extending along costal margin from base to 1/2, narrowing toward apex; third fascia obscure grayish brown, along posterior margin from 1/4 to 3/4; first blotch at 3/7, obscure grayish brown, at the costal margin; second blotch at 9/14, circular black, surrounded by brown area except for posterior margin that not reaching; third blotch ocher at 4/5; pale grayish brown streak along CuP; streak with white inside and grayish brown outside at terminal line excluding venation part; long grayish ocher fringes and short grayish brown fringes along terminal line excluding venation part. Hindwing grayish brown from costal 1/2 and posterior 3/4 connection to apex pale ocher; long grayish brown fringes and short dark grayish brown fringes at posterior from base to 3/5; short grayish brown fringes excluding venation part and long pale ocher fringes at posterior from 3/5 to apex. **Male genitalia** (Figure 8 and Figure 9). Uncus dagger-shaped, ventral margin concave at the median, apex blunt, lateral margin densely covered with setae. Gnathos slightly longer than uncus, slightly convex from base to 1/2 at the dorsal margin and slightly concave from 1/2 to apex, the ventral margin slightly concave; bar-shaped structure at the apex of the median ventral margin of gnathos, slightly shorter than half the length of gnathos with bent apex; microspines from midpoint to apex, gradually larger toward apex. Tegumen sclerotized with some long setae. Costa flat with some long setae apically. Valva subtriangular with blunt apex; cucullus as long as uncus, slightly curved, subtriangular, inner surface densely covered with short setae, ventral margin inwardly curved; harpe very weakly developed, with a few short setae throughout; sacculus wave-shaped form narrowing toward the apex, inner margin flat from base to 3/5 and concave thereafter, outer margin gently convex, apex blunt. Aedeagus 2.4 times longer than uncus, straight and gradually sclerotized toward apex; cornutus at 3/7, evenly spaced from base to near apex three pairs of spines forming a toothed wheel-like pattern with a single spine at apex, swollen vesica entirely covered with microspines inside; three tooth-shaped and one blunt spine at 5/8. **Female unknown.**

**Remarks.** The wave-shaped sacculus and inwardly curved ventral margin of the cucullus, observable through Micro-CT, were lost during the mounting process due to pressure from the cover glass. Although the complex cornutus of the aedeagus was visualized and characterized from multiple angles using Micro-CT, the internal microspines and four additional spines located at 5/8 could not be clearly resolved.

**Etymology.** This species name is a composite word derived from the Latin, ‘fulv’ and ‘corona’, meaning ‘reddish yellow’ and ‘crown’. This species name refers to the second black circular blotch and ocherous edge. In astronomy, the corona is the luminous atmosphere of celestial bodies, and it is well observed as a ring during the total solar eclipse. The species name is derived from the resemblance of the black circle and ocherous edge on the forewing to the corona seen during the solar eclipse.

**Type material.** Holotype: ♂, Indonesia, Sumatra U., Mt. Simasopa, West Sindaraya, 400 m, 28.viii. 1992, J. B. Heppner, gen. slide. No. JBNU IPE-13307/I.W. Jeong. Paratype: ♂, Indonesia, Sumatra U., Mt. Simasopa, West Sindaraya, 400 m, 28.viii. 1992, J. B. Heppner, gen. slide. No. JBNU IPE-13310/I.W. Jeong (deposited in MGCL).

**Distribution.** Indonesia: known only from the type locality.

***Casmara fuscatulipa*** **sp. nov.**

urn:lsid:zoobank.org:act:0A701D07-DF6E-4FC2-9102-E9CDA8B0D408

MorphoSource DOI of Micro-CT images: https://doi.org/10.17602/M2/M743645

MorphoSource DOI of 3D genitalia: https://doi.org/10.17602/M2/M743946

**Diagnosis.** This species is superficially similar to *C*. *agronoma*, but it has differences in the markings of the forewing. In *C*. *agronoma*, all three bundles of scales present on the dark brown forewing are brown, and at the base, there is a dark brown blotch along the costal and posterior margins, but there is no distinct blotch or fascia near the apex. But in this species, the first of three bundles of scales on the brown forewing is orange brown, while the others are reddish brown. The dark brown blotch near the base along the costal and posterior margins is edged with white. Additionally, near the apex, there is a pale reddish brown marking accompanied by a pale ocherous fascia. In the male genitalia, this species is the most similar to *C*. *agronoma*, but it has differences in the uncus, sacculus, and aedeagus. In *C*. *agronoma*, the uncus bends sharply downward at the near apex, the ventral side of the sacculus is flat, and there is a hook-shaped cornutus at 2/3 of the aedeagus. But in this species, the uncus bends slightly in the middle, the ventral side of the sacculus is convex, and there are two spine-shaped cornuti near the apex of the aedeagus.

**Description. Adult** (Figure 10). *Head*. Frons pale ocher; vertex pale ocher with pale brown at both sides; occiput pale ocher. Antenna pale ocher with cilia; scape pale brown from base to 1/2 dorsally. Labial palpus pale ocher; first segment brown dorsally; second segment irregularly mixed with brown; third segment pale brown ventrally, dark brown at 5/7. *Thorax*. Tegula brown, paler toward the apex with a white edge. Thorax brown; prothorax ocher with a white edge; cephalic margin of mesothorax dark brown; metathorax dark brown caudally. *Abdomen*. Abdomen ocher; dark brown and pale ocherous scales at each segment caudally. *Wing*. Wing expanse 35.9 mm. Forewing brown; costal grayish brown; base ocher; brown blotch with white edge along each margin of costal and posterior from base to 2/11; pale ocherous semi spear-shaped blotch along costal from 2/9 to 4/9; unclear pale reddish brown blotch along costal from 5/9 to 7/9; pale ocherous and unclear sub-diamond-shaped fascia from 3/5 to apex, not reaching terminal line; bundles of scales at 2/9, 1/2, and 2/3; first bundle orangey brown, divided in half by venation; second bundle reddish brown, divided in half by venation, white streak at base side; dark brown streak extending from base to 1/4 at center; dark brown streak from third bundle of scales narrowing gradually toward terminal line; white streak at terminal line excluding venation part; grayish brown at apex; fringes grayish brown. Hindwing brown; fringes pale brown. **Male genitalia** (Figure 11 and Figure 12). Uncus arrowhead-shaped slightly curved downward medially, apex blunt, short setae on lateral margin. Gnathos as long as uncus, tub-shaped, concave at the dorsal margin; cone-shaped structure about 1/4 length of gnathos, subtly bent at the apex, located at the apex of the ventral margin; microspines from 5/7 to apex, gradually larger toward apex. Tegumen weakly sclerotized. Costa flat with long setae apically. Valva triangular with blunt apex; cucullus slightly longer than uncus, slightly curved subtriangular with numerous short setae at the inner surface; harpe well-developed, apex densely covered with long setae; sacculus with ventral margin gently convex, ventral margin convex from base to 4/13, flat from 4/13 to 7/13, strongly concave thereafter to apex, inner margin flat from base to 3/5 and concave thereafter, outer margin angled convexly at 3/5, apex blunt, thick setae from 9/13 to apex. Aedeagus four times longer than uncus, sharply bent once; two spine-shaped cornuti at near apex, vertical, lower cornutus nearly twice the length of the other. **Female unknown.**

**Remarks.** The genitalia of this species showed almost identical features between the dissected image and the Micro-CT image.

**Etymology.** This species name is a composite word derived from the Latin, ‘fusc’ and ‘tulipa’, meaning ‘brown’ and ‘tulip’. This species name refers to a brown tulip, as the markings on the forewing collectively create a tulip shape; the dark brown blotch along the costal and posterior margins near the base resembles a leaf, the dark brown streak from the base to 1/4 resembles a stem, and the orangey brown bundle of scales resembles a flower of tulip, with the markings being predominantly brown.

**Type material.** Holotype: ♂, Taiwan, Taoyuan City, Upper Balin, Lalashan, 1500 m, 11-18.vii. 1996, J. B. Heppner, gen. slide. No. JBNU IPE-13308/I.W. Jeong (deposited in MGCL).

**Distribution.** Taiwan: known only from the type locality.

## 4. Discussion

In this study, we assessed the effectiveness of Micro-CT in species identification by examining the genitalia of three new species belonging to the genus *Casmara* through both Micro-CT and traditional dissection. Traditionally, identifying new insect species involves dissolving the abdomen in potassium hydroxide solution to separate the genitalia, then spreading and mounting them for structural observation. Although this method has been widely used, it often damages the specimen, obscures crucial morphological traits of the abdomen, and distorts the 3D characteristics of the genitalia by flattening them. Moreover, the dissection process is highly dependent on the user’s expertise, potentially damaging key diagnostic features.

To address these limitations, we employed Micro-CT to visualize the internal structures of the abdomen without causing any physical harm to the specimen. By generating 3D tomographic data, Micro-CT allows users to rotate and inspect the genitalia from multiple angles, providing a more comprehensive analysis of subtle morphological features. It also enables virtual slicing in a digital environment, allowing for the investigation of cross-sectional details that might otherwise be overlooked. In this study, we enhanced structural understanding by not only virtually separating the aedeagus but also providing detailed images of the genitalia from lateral, dorsal, ventral, and caudal views, thereby building upon previous research and offering a clearer depiction of these structures.

Nevertheless, when compared with dissected samples, Micro-CT showed limitations in capturing extremely fine structures such as setae and microspines, and it did not distinguish between sclerotized regions and the vesica. Although staining methods using iodine solutions exist to enhance tissue contrast and clarify differences between regions, these approaches can cause tissue shrinkage or information loss due to overstaining [30]. In this study, staining was not conducted because the limited number of specimens made it impossible to assess contrast variation according to staining duration. Additionally, our study did not include imaging of the female genitalia, underscoring the need for further validation regarding the technique’s ability to resolve delicate membranous structures.

Despite these challenges, this study expands the diversity of the genus *Casmara*, which has not had any new species discoveries for over a decade. However, since all specimens of these new species were collected before 1996, a thorough reassessment of the genus’s current distribution and existence is necessary to determine whether any additional or cryptic diversity remains undiscovered.

## 5. Conclusions

This study provides an example of the use of Micro-CT as a more precise method for identifying lepidopteran insect species. This method overcomes the disadvantage of damaging specimens to identify Lepidoptera species and has the strong advantage of providing new information on previously unobserved three-dimensional structures.

However, there are still technical limitations in the imaging of fine structures, and the implementation of membranous structures is not ideal. To overcome these limitations, it will likely be necessary to increase the resolution of the imaging system or enhance the contrast between tissues by staining the specimens, allowing structures to be visualized more clearly.

Nevertheless, with the present results, we can argue more strongly for distinct morphological differences in the three new species identified based on genital characteristics in two ways. We also provide raw data by presenting images captured with Micro-CT and STL files of 3D genitalia through MorphoSource DOIs, facilitating easy data utilization and enabling the 3D results to be reproduced via 3D printing.

## Figures and Tables

**Figure 1 insects-16-00747-f001:**
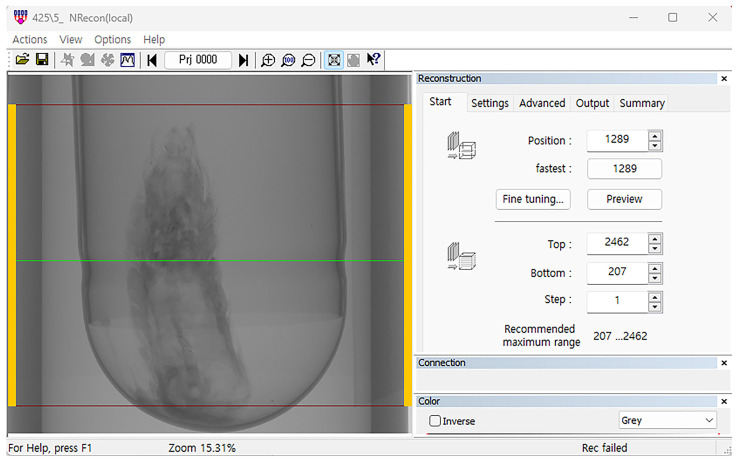
Reconstruction settings using Skyscan 1276 imaging with NRecon software. Scanning was conducted at a resolution of 4 µm using a Skyscan 1276. The region of the sample to be scanned was defined using NRecon software.

**Figure 2 insects-16-00747-f002:**
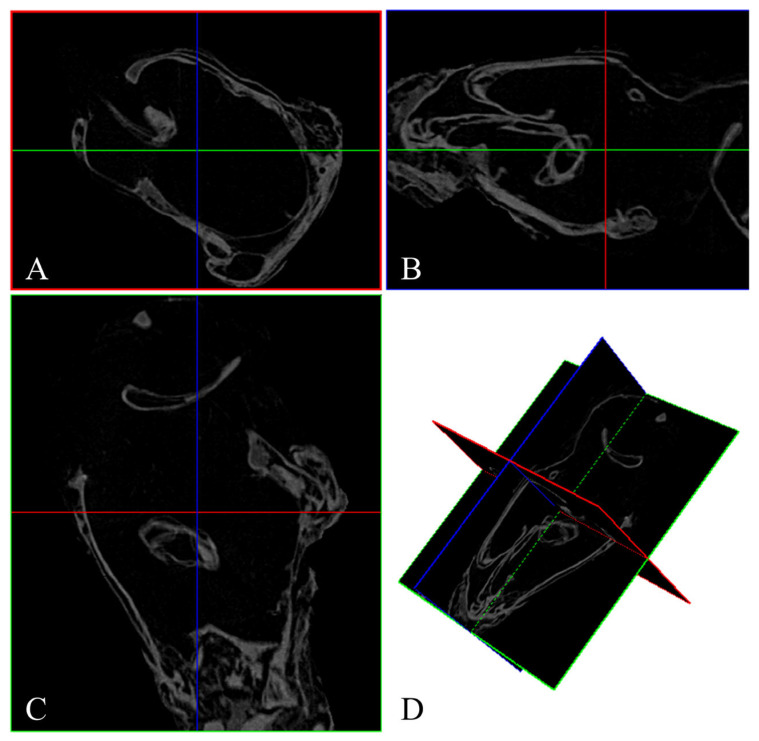
Reconstructed images by DataViewer software. (**A**) X–Y plane sliced; (**B**) Y–Z plane sliced; (**C**) Z–X plane sliced; (**D**) ortho-slice.

**Figure 3 insects-16-00747-f003:**
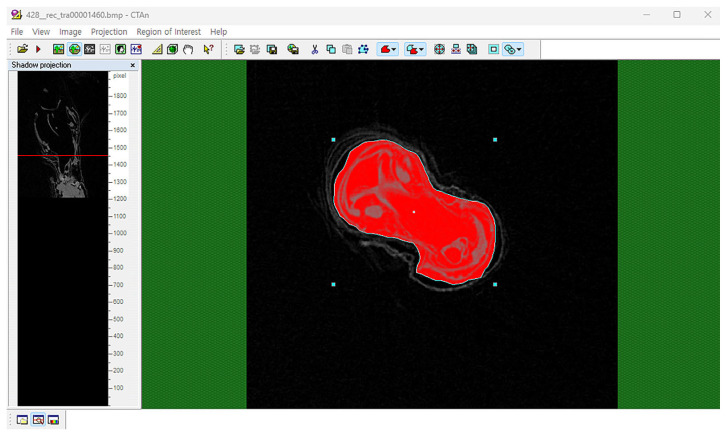
Extracting specific regions by setting ROI in CTAn. Using CTAn software, only the genitalia region was selected, and the surrounding segments were removed.

**Figure 4 insects-16-00747-f004:**
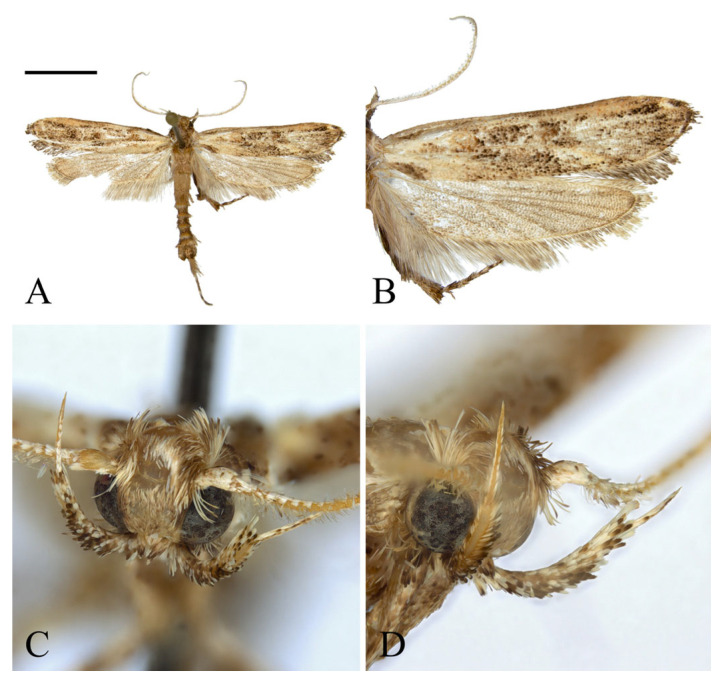
*Casmara falcatussica* **sp. nov.** (**A**) Adult; (**B**) wing; (**C**) front view of head; (**D**) lateral view of head. Scale bar: 5.0 mm.

**Figure 5 insects-16-00747-f005:**
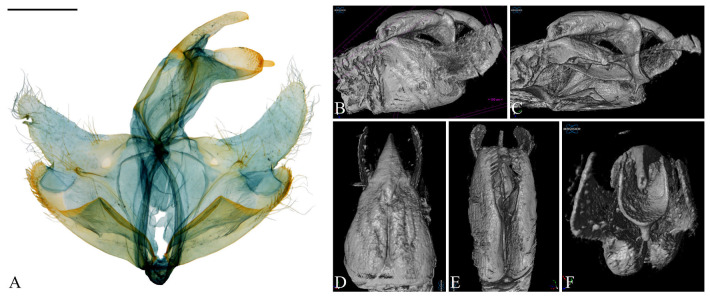
Male genitalia of *Casmara falcatussica* **sp. nov.** in 2D VS Micro-CT images. (**A**) Dissected view; (**B**) outer lateral view; (**C**) inner lateral view; (**D**) dorsal view; (**E**) ventral view; (**F**) caudal view. Scale bar: 0.5 mm.

**Figure 6 insects-16-00747-f006:**
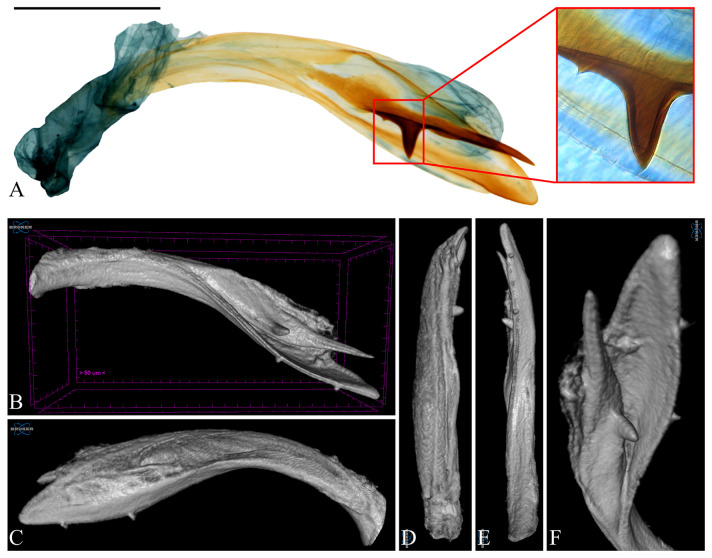
Aedeagus of *Casmara falcatussica* **sp. nov.** in 2D VS Micro-CT images. (**A**) Dissected view; (**B**) full; (**C**) lateral view; (**D**) dorsal view; (**E**) ventral view; (**F**) caudal view. Scale bar: 0.5 mm.

**Figure 7 insects-16-00747-f007:**
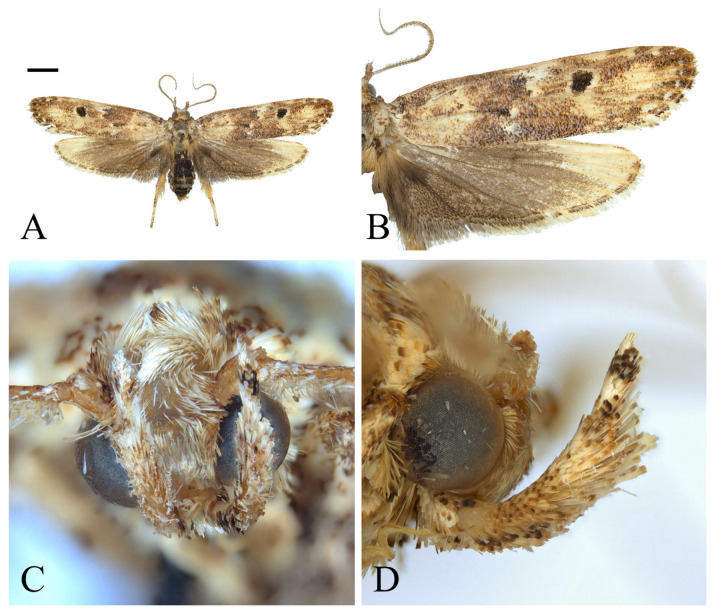
*Casmara fulvacorona* **sp. nov.** (**A**) Adult; (**B**) wing; (**C**) front view of head; (**D**) lateral view of head. Scale bar: 5.0 mm.

**Figure 8 insects-16-00747-f008:**
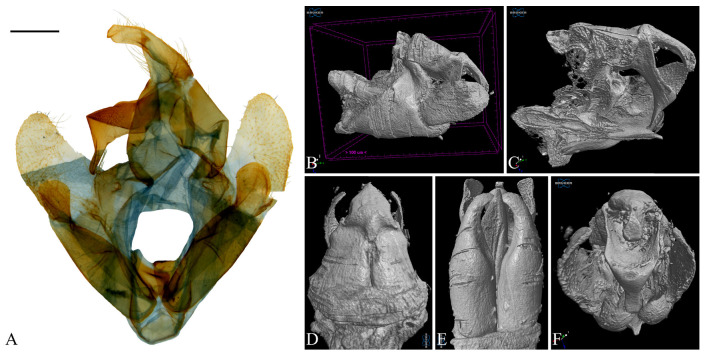
Male genitalia of *Casmara fulvacorona* **sp. nov.** in 2D VS Micro-CT images. (**A**) Dissected view; (**B**) outer lateral view; (**C**) inner lateral view; (**D**) dorsal view; (**E**) ventral view; (**F**) caudal view. Scale bar: 0.5 mm.

**Figure 9 insects-16-00747-f009:**
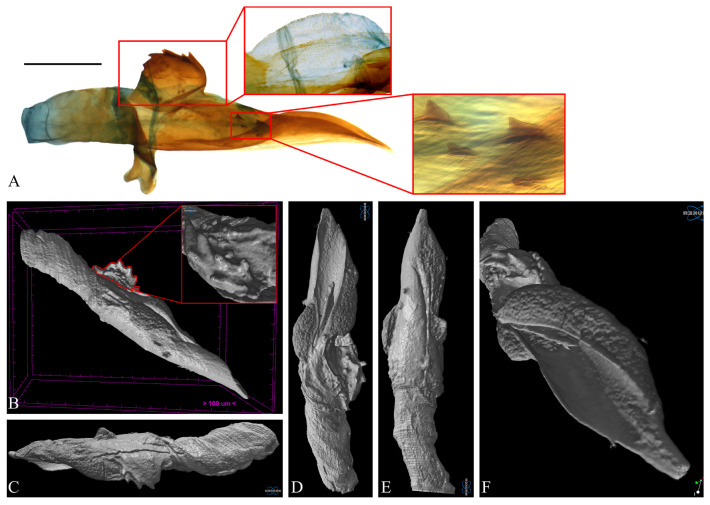
Aedeagus of *Casmara fulvacorona* **sp. nov.** in 2D VS Micro-CT images. (**A**) Dissected view; (**B**) full; (**C**) lateral view; (**D**) dorsal view; (**E**) ventral view; (**F**) caudal view. Scale bar: 0.5 mm.

**Figure 10 insects-16-00747-f010:**
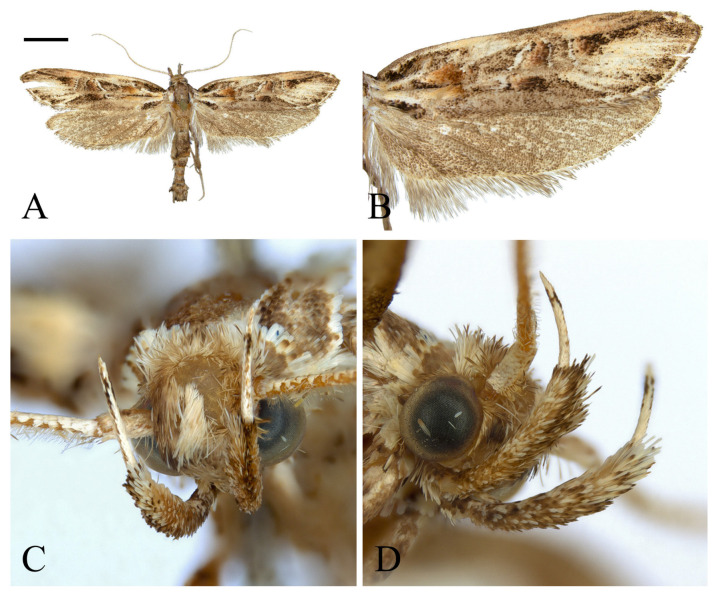
*Casmara fuscatulipa* **sp. nov.** (**A**) Adult; (**B**) wing; (**C**) front view of head; (**D**) lateral view of head. Scale bar: 5.0 mm.

**Figure 11 insects-16-00747-f011:**
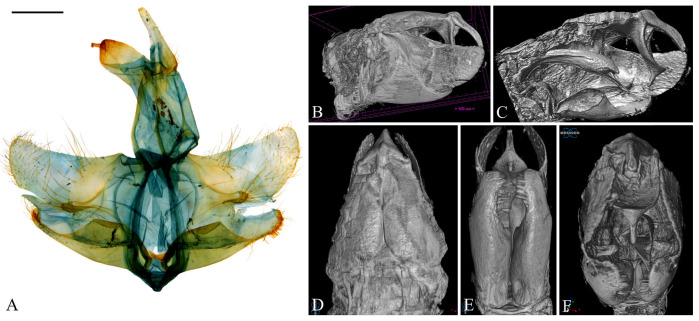
Male genitalia of *Casmara fuscatulipa* **sp. nov.** in 2D VS Micro-CT images. (**A**) Dissected view; (**B**) outer lateral view; (**C**) inner lateral view; (**D**) dorsal view; (**E**) ventral view; (**F**) caudal view. Scale bar: 0.5 mm.

**Figure 12 insects-16-00747-f012:**
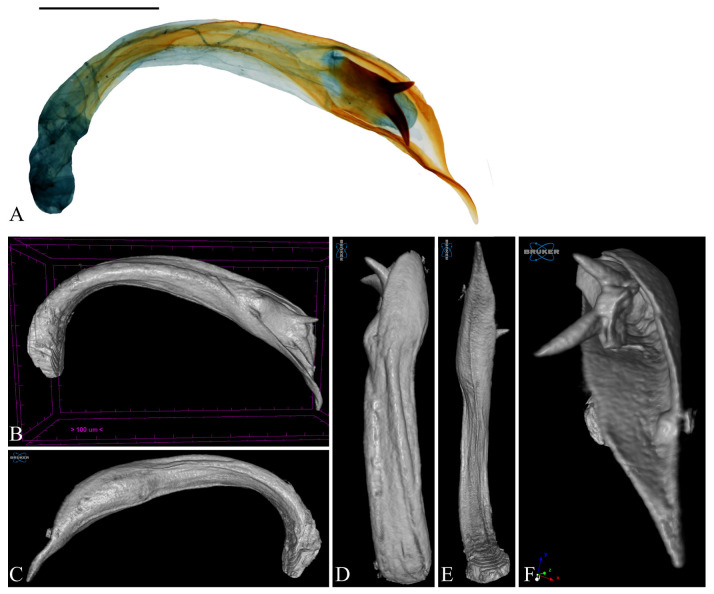
Aedeagus of *Casmara fuscatulipa* **sp. nov.** in 2D VS Micro-CT images. (**A**) Dissected view; (**B**) full; (**C**) lateral view; (**D**) dorsal view; (**E**) ventral view; (**F**) caudal view. Scale bar: 0.5 mm.

**Table 1 insects-16-00747-t001:** Checklist of the genus *Casmara* in the world. New species from this study are indicated with an asterisk *. Abbreviations: KOR, Korea; JPN, Japan; CHN, China; TWN, Taiwan; THA, Thailand; HKG, Hong Kong; VIE, Vietnam; MAS, Malaysia; INA, Indonesia; IND, India; AUS, Australia; TL, Type locality.

Species of *Casmara*	Distribution										
	KOR	JPN	CHN	TWN	THA	HKG	VIE	MAS	INA	IND	AUS
*C*. *acantha* Wang [16] Zootaxa 3239: 58–63. TL: China.			●	●							
*C*. *aduncata* Wang [16] Zootaxa 3239: 58–63. TL: China.			●								
*C*. *agronoma* Meyrick [17] Bulletin de la Section Scientifique de l’Academie Roumaine 14: 59–75. TL: China.	●	●	●	●							
*C*. *demotarcha* (Meyrick) [18] Journal of the Bombay Natural History Society 17(3): 742. TL: Khasi Hills.										●	
*C*. *diabolella* Bradley [19] Annals and magazine of natural history 12(6): 319–320. TL: Malaysia.								●			
*C*. *epicompsa* Meyrick [20] Exotic Microlepidoptera 2(17): 544. TL: Bengal, Darjiling.										●	
*C*. *exculta* (Meyrick) [21] Exotic Microlepidoptera 1(8): 237. TL: Assam, Khasis.					●		●	●	●	●	●
*C*. *falcatussica* **sp. nov.** * TL: New Taipei City.				●							
*C*. *fulvacorona* **sp. nov.** * TL: Sumatra.									●		
*C*. *fuscatulipa* **sp. nov.** * TL: Taoyuan City.				●							
*C*. *grandipennata* Moriuti [22] Tinea 12(2): 11–16. TL: Thailand.					●						
*C*. *infaustella* Walker [23] List of the specimens of lepidopterous insects in the collection of the British Museum 28: 518. TL: North Hindostan.										●	
*C*. *kalshoveni* Diakonoff [24] Tijdschrift voor Entomologie 109(3): 69. TL: Central Java.									●		
*C*. *longiclavata* Wang [16] Zootaxa 3239: 58–63. TL: China.			●	●	●	●					
*C*. *nedoshivinae* Lvovsky [14] Zoosystematic Rossica 22(1): 108. TL: Thua Thien Hue.			●				●				
*C*. *patrona* Meyrick [25] Deutsche entomologische Zeitschrift Iris 48: 38. TL: China.		●	●	●							
*C*. *phobographa* Diakonoff [24] Tijdschrift voor Entomologie 109(3): 62. TL: Sumatra.									●		
*C*. *quadrilatera* Wang [16] Zootaxa 3239: 58–63. TL: China.			●								
*C*. *regalis* Diakonoff [24] Tijdschrift voor Entomologie 109(3): 61. TL: Celebes, Pangean nr Maros.									●		●
*C*. *rhodotrachys* Diakonoff [24] Tijdschrift voor Entomologie 109(3): 67. TL: Borneo, Balikpapan.									●		
*C*. *rodochalca* Meyrick [25] Deutsche entomologische Zeitschrift Iris 48: 38. TL: China.			●								
*C*. *rufipes* Diakonoff [24] Tijdschrift voor Entomologie 109(3): 65. TL: Java.									●		
*C*. *subagronoma* Lvovsky [14] Zoosystematic Rossica 22(1): 108. TL: Vinh Phuc, Ngoc Thanh Vill.			●				●		●		
*C*. *uniata* Diakonoff [24] Tijdschrift voor Entomologie 109(3): 64. TL: Celebes, Paloe District.									●		

**Table 2 insects-16-00747-t002:** Settings for Micro-CT scanning. The system and acquisition settings for Micro-CT scanning genitalia in the genus *Casmara*.

System Settings	Acquisition Settings
Software Version = 1.6	Partial Width = 75%
Magnification Drive Version = 3.4	Image Rotation = 0.03000
Camera Drive Version = 3.4	Source Voltage (kV) = 40
Filter Drive Version = 3.4	Source Current (μA) = 100
Animal Drive Version = 3.4	Image Pixel Size (μm) = 4.0
Source Type = HAMAMATSU L10321-67	Exposure (ms) = 1200
	Filter = No Filter
Camera Type = XIMCA MH110XC-KK-TP	Camera Binning = 1 × 1
	Frame Averaging = ON (2)
Camera Pixel Size (μm) = 17.250	Scanning Trajectory = ROUND
Camera X/Y Ration = 1.0032	FF Updating Interval = 172

## Data Availability

The data generated from this study are available via DOIs on MorphoSources, with each DOI assigned to individual species.

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
