# Peer review of "A Non-Destructive Method, Micro-CT, Supports the Identification of Three New Casmara Species from Sumatra and Taiwan (Lepidoptera: Ashinagidae)†"

_insects, 2025, doi:10.3390/insects16080747_

Round 1

Reviewer 1 Report

Comments and Suggestions for Authors

The article is very well written, free of noticeable errors, and presents a compelling piece of research.

I would only suggest that, in the section where the limitations of micro-CT are discussed (line 398), a brief mention of contrast-enhancement techniques could be included. For instance, exposing samples to iodine vapours has been used to improve soft tissue contrast in micro-CT imaging. Do you think such a technique might have enhanced the resolution in this case? A short discussion (perhaps a couple of lines) on the potential benefits or drawbacks of contrast agents such as iodine, phosphotungstic acid or osmium tetroxide could strengthen this section and offer useful context for readers unfamiliar with these approaches.

Author Response

  1. I would only suggest that, in the section where the limitations of micro-CT are discussed (line 398), a brief mention of contrast-enhancement techniques could be included. For instance, exposing samples to iodine vapours has been used to improve soft tissue contrast in micro-CT imaging. Do you think such a technique might have enhanced the resolution in this case? A short discussion (perhaps a couple of lines) on the potential benefits or drawbacks of contrast agents such as iodine, phosphotungstic acid or osmium tetroxide could strengthen this section and offer useful context for readers unfamiliar with these approaches.

ANSWER> Thank you for your feedback. As you suggested, staining methods to enhance contrast are indeed crucial analytical techniques for micro-CT imaging; thus, we have included additional information about iodine staining and its potential drawbacks. Furthermore, we clarified that staining was not performed in this study due to the limited number of specimens, which prevented us from assessing contrast variation according to staining duration. The supplemented content is as follows:

<Lines 399–403> Although staining methods using iodine solutions exist to enhance tissue contrast and clarify differences between regions, these approaches can cause tissue shrinkage or information loss due to overstaining. In this study, staining was not conducted because the limited number of specimens made it impossible to assess contrast variation according to staining duration.

Reviewer 2 Report

Comments and Suggestions for Authors

1. The genus Casmara belongs to the family Oecophoridae. Members of Casmara are much different from those of Stathmopodidae in size and morphological characters. Reclassification of its status from Oecophoridae to Stathmopodidae should be based on a comprehensive study rather than through a molecular phylogenetic study of limited genera and species. This is only a suggestion for authors to consider.

2. Casmara falcatussica sp. nov.

Same species as C. longiclavata S. Wang, 2012 that is also distributed in Chinese Taiwan. If the authors treat it as a new species, it is better to compare the male genital characters of the new species with those of C. longiclavata.

The authors compared the wing markings of the new species with those of C. subagronoma Lvovsky. The forewing markings of the new species are abraded, and have they checked the type of C. subagronoma?

3. Casmara fulvacorona sp. nov.

Same species as C. aduncata S. Wang, 2012 that was described from Yunnan, China. If the authors treat it as a new species, they are suggested to compare the male genital characters of the new species with those of C. aduncata.

How could it be similar to C. phobographa Diakonoff? Quite different in appearance and male genitalia.

The authors compared the male genitalia of the new species with those of C. agronoma. However, the two species are greatly different in superficial characters. There is no need to compare a new species with a species that is much easy to distinguish superficially. Please compare with similar species for the purpose of species identification.

4. Female genitalia unknown’ is a wrong expression. Correct expression is ‘Female unknown’ since there is no female specimen at all.

Author Response

  1. The genus Casmarabelongs to the family Oecophoridae. Members of Casmaraare much different from those of Stathmopodidae in size and morphological characters. Reclassification of its status from Oecophoridae to Stathmopodidae should be based on a comprehensive study rather than through a molecular phylogenetic study of limited genera and species. This is only a suggestion for authors to consider.

ANSWER> Based on molecular phylogenetic studies, the genus Casmara was reclassified from Oecophoridae to Stathmopodidae in 2016 and subsequently classified to Ashinagidae by Wang & Li in 2020; accordingly, I have revised and supplemented the title and relevant sections of the introduction. Thank you for your review.

  1. Casmara falcatussicasp. nov.

Same species as C. longiclavata S. Wang, 2012 that is also distributed in Chinese Taiwan. If the authors treat it as a new species, it is better to compare the male genital characters of the new species with those of C. longiclavata.

The authors compared the wing markings of the new species with those of C. subagronoma Lvovsky. The forewing markings of the new species are abraded, and have they checked the type of C. subagronoma?

ANSWER> As suggested, the diagnosis of this new species was revised by comparing it with the described species C. longiclavata Wang, 2012. Diagnostic differences were observed in the cucullus and aedeagus. The previously written diagnostic comparison between this new species and C. subagronoma has been removed. <Lines 159–177>

  1. Casmara fulvacorona sp. nov.

Same species as C. aduncata S. Wang, 2012 that was described from Yunnan, China. If the authors treat it as a new species, they are suggested to compare the male genital characters of the new species with those of C. aduncata.

How could it be similar to C. phobographa Diakonoff? Quite different in appearance and male genitalia.

The authors compared the male genitalia of the new species with those of C. agronoma. However, the two species are greatly different in superficial characters. There is no need to compare a new species with a species that is much easy to distinguish superficially. Please compare with similar species for the purpose of species identification.

ANSWER> Thank you for your review. As per your suggestion, the diagnosis has been revised to compare the new species with the morphologically similar C. aduncata. Diagnostic differences were found in the structure near the apex of the gnathos and in the aedeagus. Additionally, a comparison with C. exculta, a similar Casmara species recorded from the same country, has been included to facilitate species identification in the region. <Lines 234–243>

  1. Female genitalia unknown’ is a wrong expression. Correct expression is ‘Female unknown’ since there is no female specimen at all.

ANSWER> Thank you for your review. I have revised all instances previously written as “Female genitalia. Unknown.” To “Female unknown”.

Reviewer 3 Report

Comments and Suggestions for Authors

For Editors and Authors,

This study applied an updated technique, Micro-CT, to address the issue of character loss and abdominal characters destruction that often occurs when Lepidopterists traditionally dissect and examine genitalia. Although the results are still at a preliminary stage, this method has great potential to elevate taxonomic research to a new level, especially if current limitations (e.g., resolution, differentiation between membranous and sclerotized structures) can be overcome in the future.

However, despite the novelty, there are some major issues that must have been checked and corrected in this study.

  1. The taxonomic status of the genus Casmara.

According to the most updated phylogenetic study of Gelechioidea (Wang & Li, 2020; DOI: https://doi.org/10.1111/zsc.12407), the genus Casmara has been assigned to a newly established family, Ashinagidae, based on more comprehensive taxon sampling. While there may be differing viewpoints regarding this taxonomic placement, it is supported by both morphological and molecular evidence, which I find reasonable.

The earlier clustering of Casmara with Stathmopodidae in Kim et al. (2016) was likely due to the absence of Ashinaga from the sampling. Notably, the phylogenetic position of Casmara remains consistent between Kim et al. (2016) and Wang & Li (2020), indicating that the discrepancy stems from sampling differences rather than conflicting data. Moreover, Casmara exhibits substantial morphological differences from typical stathmopodid moths, further justifying its separation from Stathmopodidae.

Since the manuscript refers to the family Stathmopodidae in several sections (Lines 29–30, 48–54, 76–79), these parts should be revised accordingly. Alternatively, if the authors choose to retain Casmara within Stathmopodidae, a clear justification for this taxonomic decision should be provided.

  1. The condition of the type materials.

Conventionally, multiple specimens should be examined before naming a new species, as individual variation can occur. However, in this study, two of the three newly described species are based on only a single specimen. (If additional specimens exist, please ensure they are listed in the type material section.)

Additionally, many of the type specimens appear to be in poor condition, with wing patterns that are faint or indistinct. As such, diagnoses based primarily on wing patterns are, in my view, unreliable. This concern is particularly relevant to C. falcatussica and C. fulvacorona.

For C. falcatussica, the holotype is extremely faded, and reliable diagnostic features are largely limited to the genitalia. However, the hook-shaped cornutus and the overall structure of the male genitalia closely resemble those of C. acantha Wang, Zhang & Wang, 2012. Could the differences in wing markings simply be due to the poor condition of the specimen?

In the case of C. fulvacorona, although it possesses a distinctive cornutus that is uncommon in this genus, this feature is also very similar to that of C. aduncata Wang, Zhang & Wang, 2012. The diagnostic distinction between these two species is based solely on wing coloration and pattern. I question whether they are truly separate species. At a minimum, the authors should include a comparative analysis of the genitalia of these two species in the diagnosis section to support their taxonomic conclusions.

  1. The recorded species in Taiwan

According to the Catalogue of Life in Taiwan (TaiCoL), four Casmara species have been recorded: C. acantha Wang, Zhang & Wang, 2012; C. agronoma Meyrick, 1931; C. longiclavata Wang, Zhang & Wang, 2012; and C. patrona Meyrick, 1925. Since this database is published and directly linked to GBIF, I recommend that the authors incorporate this information into Table 1 and cross-check the current species list to ensure there are no omissions from the online database.

Additionally, to facilitate future local research, it would be beneficial to include comparative diagnoses of the newly described species alongside the previously recorded Taiwanese and Indonesian species within each diagnosis section.

Minor Comments:

Title: changing the “em-dash” into “comma”: …method, Micro-CT, …  

Abstract: Line 39–42: (Micro-CT enabled … enhancing morphological analysis) moves up to the end of Line 34–35: (…through Micro-CT and traditional dissection.)

Materials and Methods: List out the following systems (references) of the classification system, terminology for genitalia, and wing patterns.

Materials and Methods: Line 131 typo. 2.4.3 -> 2.4

Materials and Methods: section 2.4. I suggest combining the subsections into a single section. As they are now, the subsections resemble software documentation rather than the typical writing style of a Materials and Methods section.

Results: Line 169–171: I’m not sure what you’re trying to convey in this sentence. Please double-check the grammar and clarify the meaning.

Results: Line 224: Taipei City -> New Taipei City.

Results: Line 375: Palin -> Balin

Discussion: Line 382: Previously -> Traditionally

Author Response

However, despite the novelty, there are some major issues that must have been checked and corrected in this study.

  1. The taxonomic status of the genus Casmara.

According to the most updated phylogenetic study of Gelechioidea (Wang & Li, 2020; DOI: https://doi.org/10.1111/zsc.12407), the genus Casmara has been assigned to a newly established family, Ashinagidae, based on more comprehensive taxon sampling. While there may be differing viewpoints regarding this taxonomic placement, it is supported by both morphological and molecular evidence, which I find reasonable.

The earlier clustering of Casmara with Stathmopodidae in Kim et al. (2016) was likely due to the absence of Ashinaga from the sampling. Notably, the phylogenetic position of Casmara remains consistent between Kim et al. (2016) and Wang & Li (2020), indicating that the discrepancy stems from sampling differences rather than conflicting data. Moreover, Casmara exhibits substantial morphological differences from typical stathmopodid moths, further justifying its separation from Stathmopodidae.

Since the manuscript refers to the family Stathmopodidae in several sections (Lines 29–30, 48–54, 76–79), these parts should be revised accordingly. Alternatively, if the authors choose to retain Casmara within Stathmopodidae, a clear justification for this taxonomic decision should be provided.

ANSWER> Thank you for your feedback. As suggested, I have incorporated the results of the 2020 molecular phylogenetic study by Wang & Li, reclassifying the genus Casmara into Ashinagidae. Accordingly, I have revised the title, keywords, and introduction.

  1. The condition of the type materials.

Conventionally, multiple specimens should be examined before naming a new species, as individual variation can occur. However, in this study, two of the three newly described species are based on only a single specimen. (If additional specimens exist, please ensure they are listed in the type material section.)

Additionally, many of the type specimens appear to be in poor condition, with wing patterns that are faint or indistinct. As such, diagnoses based primarily on wing patterns are, in my view, unreliable. This concern is particularly relevant to Cfalcatussica and Cfulvacorona.

For Cfalcatussica, the holotype is extremely faded, and reliable diagnostic features are largely limited to the genitalia. However, the hook-shaped cornutus and the overall structure of the male genitalia closely resemble those of Cacantha Wang, Zhang & Wang, 2012. Could the differences in wing markings simply be due to the poor condition of the specimen?

In the case of Cfulvacorona, although it possesses a distinctive cornutus that is uncommon in this genus, this feature is also very similar to that of Caduncata Wang, Zhang & Wang, 2012. The diagnostic distinction between these two species is based solely on wing coloration and pattern. I question whether they are truly separate species. At a minimum, the authors should include a comparative analysis of the genitalia of these two species in the diagnosis section to support their taxonomic conclusions.

ANSWER> Thank you for your thorough review. We fully agree with your concerns; however, regrettably, no additional specimens beyond those listed in the type material are available, limiting further verification of morphological variation. Nevertheless, we include a detailed diagnostic comparison of the male genitalia for the species you indicated, emphasizing the characters that distinguish them from previously recorded taxa.

We have added a diagnostic comparison between C. falcatussica sp. nov. and C. acantha, highlighting the morphological differences observed in the cucullus and cornutus. <Lines 167–172>

We have also added a diagnostic comparison between C. fulavacorona and C. aduncata, clearly specifying the morphological differences in the cucullus and aedeagus. <Lines 229–238>

  1. The recorded species in Taiwan

According to the Catalogue of Life in Taiwan (TaiCoL), four Casmara species have been recorded: Cacantha Wang, Zhang & Wang, 2012; Cagronoma Meyrick, 1931; Clongiclavata Wang, Zhang & Wang, 2012; and Cpatrona Meyrick, 1925. Since this database is published and directly linked to GBIF, I recommend that the authors incorporate this information into Table 1 and cross-check the current species list to ensure there are no omissions from the online database.

Additionally, to facilitate future local research, it would be beneficial to include comparative diagnoses of the newly described species alongside the previously recorded Taiwanese and Indonesian species within each diagnosis section.

ANSWER> Thank you for your review. As you suggested, I have supplemented Table 1 with the distribution of Casmara species using data from GBIF. This has added distributions in Hong Kong and Australia, which were not previously covered. Additionally, as suggested, I have included diagnostic comparisons between each new species and previously recorded Casmara species from the same countries to clarify species identification within the same region.

Minor Comments:

Title: changing the “em-dash” into “comma”: …method, Micro-CT, …  

ANSWER> I have revised it. Thank you.

Abstract: Line 39–42: (Micro-CT enabled … enhancing morphological analysis) moves up to the end of Line 34–35: (…through Micro-CT and traditional dissection.)

ANSWER> As suggested, I have rearranged the sentence order and revised the paragraph accordingly. Thank you for your recommendation.

Materials and Methods: List out the following systems (references) of the classification system, terminology for genitalia, and wing patterns.

ANSWER> Thank you for your valuable review. As you suggested, I have added a description of the classification system of Casmara and clarified that the terminology follows Heppner’s references.

<Lines 102, 150–153>

Materials and Methods: Line 131 typo. 2.4.3 -> 2.4          

ANSWER> I have reviewed and revised it. Thank you for checking.

Materials and Methods: section 2.4. I suggest combining the subsections into a single section. As they are now, the subsections resemble software documentation rather than the typical writing style of a Materials and Methods section.

ANSWER> As suggested, the three former subheadings in Materials and Methods, section 2.4, “3D reconstruction of genitalia”, have been consolidated into a single, concise, and clear paragraph. Thank you for the recommendation.

Results: Line 169–171: I’m not sure what you’re trying to convey in this sentence. Please double-check the grammar and clarify the meaning.

ANSWER> The section you indicated has been removed during revision, and the diagnosis has been rewritten by comparing Casmara falcatussica sp. nov. with a species that is morphologically more similar. We apologize for any confusion this may have caused.

Results: Line 224: Taipei City -> New Taipei City.

Results: Line 375: Palin -> Balin

Discussion: Line 382: Previously -> Traditionally

ANSWER> All three sections you mentioned have been revised as suggested. Thank you for your review.

Reviewer 4 Report

Comments and Suggestions for Authors

The inclusion of the genus Casmara in the family Stathmopodidae is based on a single article. This action is highly questionable and is not accepted by most microlepidopterists. The new species surely belong to Casmara, and their descriptions are justified, but the family affiliation has to be corroborated and reaffirmed.

Author Response

The inclusion of the genus Casmara in the family Stathmopodidae is based on a single article. This action is highly questionable and is not accepted by most microlepidopterists. The new species surely belong to Casmara, and their descriptions are justified, but the family affiliation has to be corroborated and reaffirmed.

ANSWER> Thank you for your comments. Reflecting your suggestions, I have added that, based on molecular phylogenetic studies, the genus Casmara was classified into Stathmopodidae in 2016 and subsequently reclassified into Ashinagidae by Wang & Li in 2020. Accordingly, I have revised the title and parts of the introduction.

Round 2

Reviewer 3 Report

Comments and Suggestions for Authors

This revised version of the manuscript has addressed my previous comments, and I find that both the fluency and overall quality have improved significantly. I believe it is now suitable for acceptance.

I only noticed one minor issue that still requires revision:

Line 151: The author(s) of the family Ashinagidae were not provided. Please include this information.

Aside from that, I am satisfied with the rest of the manuscript and am pleased to see this research moving toward publication.

Author Response

This revised version of the manuscript has addressed my previous comments, and I find that both the fluency and overall quality have improved significantly. I believe it is now suitable for acceptance.

I only noticed one minor issue that still requires revision:

Line 151: The author(s) of the family Ashinagidae were not provided. Please include this information.

Aside from that, I am satisfied with the rest of the manuscript and am pleased to see this research moving toward publication.

ANSWER> Thank you for your valuable feedback. As per your suggestions, the previously written “Family Ashinagidae” has been corrected to “Family Ashinagidae Matsumura, 1929”.

Thank you for taking the time to review this manuscript. Your thoughtful feedback is sincerely appreciated.

Reviewer 4 Report

Comments and Suggestions for Authors

I found my criticism as accepted and the text was corrected accordingly.

Now, it is a good paper.

Author Response

I found my criticism as accepted and the text was corrected accordingly.

Now, it is a good paper.

ANSWER> I sincerely appreciate you taking the time to review this manuscript despite your busy schedule.